# Understanding voltage decay in lithium-excess layered cathode materials through oxygen-centred structural arrangement

Seungjun Myeong [1], Woongrae Cho [1], Wooyoung Jin [1], Jaeseong Hwang[1], Moonsu Yoon[1], Youngshin Yoo[1], Gyutae Nam[1], Haeseong Jang [1], Jung-Gu Han[1], Nam-Soon Choi[1], Min Gyu Kim[2] & Jaephil Cho[1]

Lithium-excess 3d-transition-metal layered oxides ($Li_{1+x}Ni_yCo_zMn_{1-x-y-z}O_2$, >250 mAh g$^{-1}$) suffer from severe voltage decay upon cycling, which decreases energy density and hinders further research and development. Nevertheless, the lack of understanding on chemical and structural uniqueness of the material prevents the interpretation of internal degradation chemistry. Here, we discover a fundamental reason of the voltage decay phenomenon by comparing ordered and cation-disordered materials with a combination of X-ray absorption spectroscopy and transmission electron microscopy studies. The cation arrangement determines the transition metal-oxygen covalency and structural reversibility related to voltage decay. The identification of structural arrangement with de-lithiated oxygen-centred octahedron and interactions between octahedrons affecting the oxygen stability and transition metal mobility of layered oxide provides the insight into the degradation chemistry of cathode materials and a way to develop high-energy density electrodes.

[1] Department of Energy Engineering and School of Energy and Chemical Engineering, Ulsan National Institute of Science and Technology (UNIST), Ulsan 689-798, South Korea. [2] Beamline Research Division, Pohang Accelerator Laboratory (PAL), Pohang, Kyungbuk 37673, Republic of Korea. Correspondence and requests for materials should be addressed to M.G.K. (email: mgkim@postech.ac.kr) or to J.C. (email: jpcho@unist.ac.kr)

Despite the success of rechargeable lithium-ion batteries in Information Technology (IT) devices, many challenges still remain for large scale energy storage systems such as electric vehicles (EV), especially in terms of high-energy density and low cost[1–3]. High-capacity Li-excess 3d-transition-metal (3dTM) layered oxides ($Li_{1+x}Mn_yNi_zCo_{1-x-y-z}O_2$) are considered as a realistic candidate material for the approaching battery era owing to their high reversible capacities exceeding 250 mAh g$^{-1}$ and cost competitiveness because of inexpensive manganese[4,5]. Nevertheless, industrial application is challenging due to their poor rate capability and severe discharge voltage decay[6–11]. Among these drawbacks, voltage decay phenomenon, which is not observed in other layered oxides, is an unique and delicate issue for researchers. Previous studies with cation-focused analysis interpreted this phenomenon as 'layered-to-spinel' phase transition and suggested 'surface doping' as a solution for the voltage decay[12–25]. Furthermore, 4d and 5d transition-metal (TM) oxides having a pure C2/m monoclinic structure, such as $Li_2Ru_{1-y}M_yO_3$ (M = Ti, Mn, Sn), were introduced to reveal origin of the voltage decay[26].

However, Li-excess materials with 3dTM show structural uniqueness ($R\bar{3}m + C2/m$) and no true $O_2^{2-}$ species on oxide oxidation, which is different from those of the heavier TM oxides[27,28]. Recent important studies on Li-excess materials mainly focused on elucidating oxygen redox chemistry based on oxygen-centred octahedron ($M_6O$)[28–30]. For instance, Ceder et al. suggested a useful participation of new Li–O–Li configuration in the oxygen redox activity for various Li-excess materials. Bruce et al. described electron-hole localization on oxygen surrounded by $Mn^{4+}/Li^+$ cations ($O–Mn^{4+}/Li^+$) to explain the oxygen chemistry in $Li_{1.2}Ni_{0.13}Co_{0.13}Mn_{0.54}O_2$. Despite such a reasonable local structural approaches, the voltage decay phenomenon can be still open issue, owing to a lack of consideration for long-range arrangement of de-lithiated $M_6O$ (M = transition metal or Li cation) octahedron and interactions affecting the overall oxygen stability. Therefore, multilateral and macroscopic analysis reflecting oxygen characteristics in a long-range structural arrangement is required to understand the voltage decay.

In this study, we focus on ordered and cation-disordered Li-excess 3dTM layered oxides showing significantly different degree of voltage decay. Operando atomic-selective X-ray absorption spectroscopic (XAS) analysis was conducted by synchronizing with dQ/dV plots to obtain a direct evidence of the voltage decay phenomenon. By combining spectroscopic and microscopic techniques, we observed the different gradation in TM-O covalency and atomic rearrangement considering structural uniqueness. Furthermore, different variation in specific TM-O bonding and TM ion migration depending on cycles, which can be a crucial parameter determining the degree of voltage decay, were revealed during cycling. Furthermore, we identify long-range structural arrangement with three types of de-lithiated oxygen-centred octahedron ($M_4O$) to discover a fundamental reason how atomic arrangement affects structural stability of Li-excess 3dTM layered oxides and the degree of voltage decay.

## Results

**Voltage decay phenomenon during cycling.** The Li-excess 3dTM layered oxides with different TM compositions, '$Li_{1.15}Mn_{0.51}Co_{0.17}Ni_{0.17}O_2$ composition with well-ordered layered phase and long-range ordered Li-TM-TM arrangement (denoted as O-MNC) and $Li_{1.09}Mn_{0.55}Ni_{0.32}Co_{0.043}O_2$ composition with cation-disordered layered phase and short-range ordered Li–TM–TM arrangement (denoted as D-MNC) were prepared by co-precipitation method. Morphology and crystal structure were confirmed by scanning electron microscopy (SEM)

and X-ray diffraction (XRD) (Supplementary Figs. 1 and 2). Furthermore, crystallographic parameters for O-MNC and D-MNC were obtained by Rietveld refinement (Supplementary Tables 1 and 2) and discussion in Supplementary Note 1. The results indicate that ~3 times more Ni and Co ions have occupied the Li site in D-MNC (total 0.10 mol) compared to O-MNC (total 0.03 mol). O-MNC and D-MNC show initial discharge capacities over 250 mAh g$^{-1}$ with coulombic efficiency over 90%. For an exact comparison of voltage decay, we considered the 1st cycle reversible capacity because the amount of extracted lithium ion from the cathode material is directly related to structural degradation, and thus, voltage decay. Long-term cycling test of O-MNC were conducted with two different voltage cutoff conditions, 2.00–4.35 V (denoted as OL-MNC) and 2.00–4.60 V (denoted as OH-MNC) with 0.5 C and 1.0 C charge and discharge C-rates, respectively. Thereafter, we confirmed the discharge capacity from the 1st cycle: 150 mAh g$^{-1}$ for OL-MNC and 200 mAh g$^{-1}$ for OH-MNC. In addition, the D-MNC half-cell exhibited 200 mAh g$^{-1}$ at the 1st cycle in the voltage window of 2.00–4.60 V. All samples showed capacity retention of almost 93 % after 100 cycles (Supplementary Fig. 3 and Supplementary Table 3). The coloured regions in Fig. 1a, b show the difference in the degree of voltage decay from the 10th to 100th cycle for samples. Voltage decay was observed in both on charge and discharge process. The average discharge voltages of OL-MNC and OH-MNC decreased by 31.6 mV and 317.0 mV during cycling, respectively (Fig. 1a); however, D-MNC shows only 56.0 mV of voltage decay (Fig. 1b and Supplementary Table 4). This tendency is definitely recognized with dQ/dV plots shown in Fig. 1c. OH-MNC significantly exhibited both a disappearance of reduction peak at around 3.70 V and the new development of reduction peak at around 2.80 V during cycling, while OL-MNC retained its original profile even after 100th cycle. In case of D-MNC, the overall change in reduction peaks is suppressed despite the same electrochemical testing condition with OH-MNC.

**Comparison of redox mechanism variation.** In order to elucidate the different redox reactions observed in dQ/dV plot during cycling, operando Mn, Ni, and Co K-edges X-ray absorption near edge structure (XANES) spectra were collected for the 10th and 100th electrodes using the on-the-fly scan mode under the same electrochemical test condition above (Fig. 2a–d). The oxidation states of TMs in initial, charged, and discharged states at the 10th and 100th cycle were obtained from least-square method (Supplementary Fig. 4–6 and Supplementary Table 5)[31,32]. At the 10th cycle, OH-MNC shows a wide variation range of oxidation state ($\Delta$Ox) and reduction state ($\Delta$Red) towards higher oxidation states at the charged state due to a high cutoff voltage ($Mn^{3.65+}/Ni^{3.08+}/Co^{2.98+}$). In particularly, irreversible redox behaviour is observed at Mn state ($\Delta$Ox: 0.46/$\Delta$Red: 0.35). This contrasts with the redox reactions of D-MNC, which resulted in narrow $\Delta$Ox/$\Delta$Red, lower oxidation states ($Mn^{3.53+}/Ni^{2.88+}/Co^{2.85+}$), and relatively reversible Mn redox ($\Delta$Ox: 0.34/$\Delta$Red: 0.29). This feature indicates that charged TM ion in D-MNC form a relatively covalent bond with surrounded oxygen compare to OH-MNC because low oxidation number in TM ion is arisen from existence of numbers of shared electrons in TM-O bonding. During the 100th cycle, wider $\Delta$Ox/$\Delta$Red was commonly observed in all TMs of D-MNC however, almost same oxidation states of Mn ion at 100th charged states with 10th charged states were observed in both OH-MNC and D-MNC.

In order to elucidate real-time redox reaction depending on the voltage decay during 10th and 100th cycle, differentiated XANES spectra ($X_i$ - $X_{pristine}$) were presented with 2D-contour map by synchronizing with dQ/dV plots in Fig. 2a–d and Supplementary

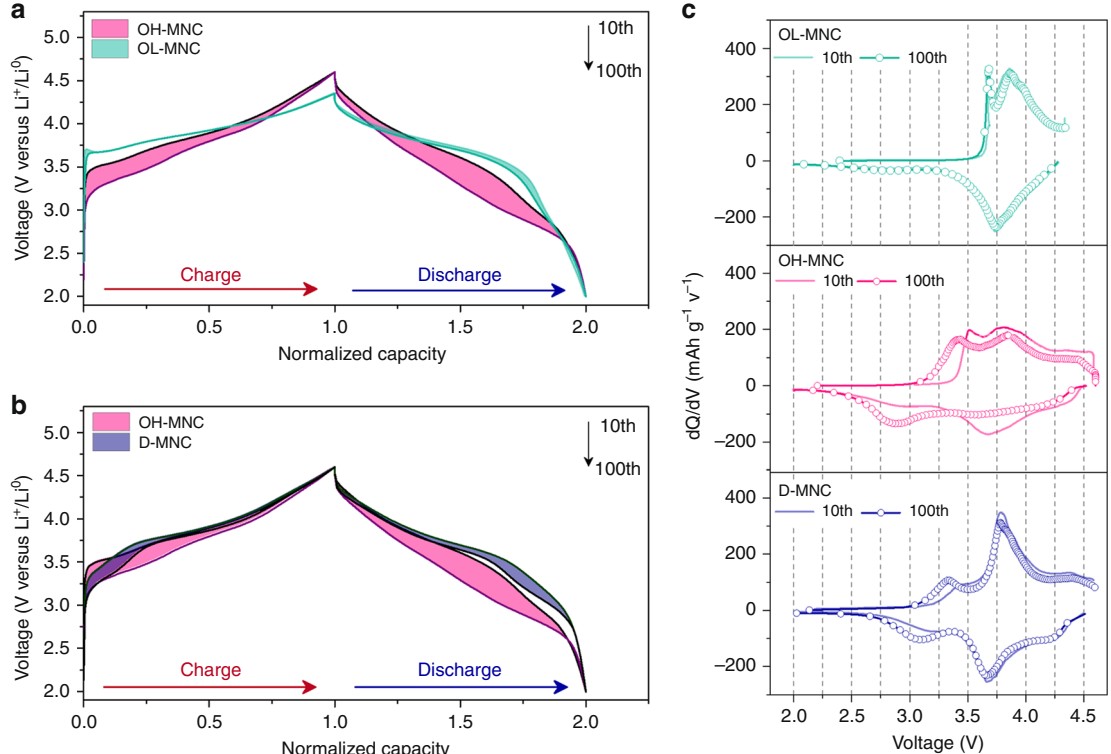

**Fig. 1** Cycle behaviuor of O-MNC and D-MNC with a view towards voltage decay. Normalized voltage profile variation for O-MNC and D-MNC from 10th to 100th cycles; Voltage profile variation of **a** O-MNC in the 2.00–4.35 V and 2.00–4.60 V (vs. Li-metal) potential region. **b** O-MNC and D-MNC in the 2.00–4.60 V (vs. Li-metal) potential region; 0.5 C-rate charge 1.0 C-rate discharge condition. The capacity normalization was performed by taking the maximum capacity in the corresponding cycle as unity. **c** dQ/dV plots for the O-MNC and D-MNC obtained from voltage profiles in **a**, **b**

Fig. 7. In the crossline intensities of each peak during the 10th cycle, Mn K-edge XANES of OH-MNC is significantly more irreversible than that of D-MNC (Fig. 2e, f), while reversible behaviour of Ni and Co is observed in both OH-MNC and D-MNC (Supplementary Figs. 8 and 9).

During the 100th cycle, D-MNC not only retains the overall patterns of TM K-edges, but also shows more solidified peak features (Fig. 2d). However, the 100th OH-MNC shows different 2D contour map pattern from 10th cycle (Fig. 2c) in which overall peak intensity of patterns are gradually diminished at Ni and Co K-edge. Furthermore, Mn K-edge of 100th cycle shows very different overall peak features contrast to that of 10th cycle. In the 100th charge process of OH-MNC, the suppressed oxidation reaction of Ni and Co starts at higher voltage and the active oxidation reaction of Mn occurs at lower voltage compared with D-MNC in the 100th charging process, which are well matched with the tendency of dQ/dV plot. Consequently, it can be expected that the structural instability associated with irreversible Mn ion redox at 10th cycle is a critical factor causing the abnormal behaviour in redox activities of the Ni/Co and Mn ions during cycling, resulting in the noticeable peak decrease at around 3.70 V and peak increase at around 2.80 V in dQ/dV plot.

**Electronic structure of TM-O bonding.** The electron-hole state in TM-O bonding is closely related to structural stability, resulting in TM redox activity variation; thus, scanning transmission X-ray microscopy (STXM) was performed to reveal the oxygen state in materials, which undergo both TM and oxygen redox during electrochemical reactions. Figure 3a–c shows the O K-edge soft X-ray absorption spectroscopy (Soft-XAS) data of free standing pristine and 10th and 100th cycled (discharged state) OL-MNC, OH-MNC, and D-MNC single particles chosen by microscopy imaging (Supplementary Fig. 10). Firstly, the broad main peak of the O K-edge spectra above 534.0 eV corresponds to transition from O1s to the hybridized state of O2p and TM4sp orbitals. Secondly, the shaded area below 534.0 eV in Fig. 3 indicates the hybridized state of O2p and TM3d orbitals (O2p–TM3d). The distinct doublet peak feature in the pre-edge region (peak A at ~529.5 eV and peak B ~531.0 eV) are associated with the transition of O1s electron to the hybridized state of O2p with TM3d$t_{2g}$ and TM3d$e_g$ states, respectively[28,33,34].

The intensity of peak A in the pristine O-MNC is remarkably lower than that in D-MNC[35], implying that the electrons originating from oxygen vacancy ($O_O = V''_O + 2e^- + 0.5O_2$) occupy the TM3d$t_{2g}$ states. Therefore, higher intensity in D-MNC means more covalent interaction on O2p–TM3d$t_{2g}$ hybridized orbital (Hybridization is a model that modifies atomic orbitals to explain the covalent bonding phenomenon 'covalancy'). After the 10th and 100th cycles, the integrated intensity of the pre-edge peak (area of the shaded region in Fig. 3a), which is directly related to the degree of TM-O covalency, gradually decreased by ~0.82/~2.62% for OL-MNC and dramatically decreased by ~3.54/~16.22% for OH-MNC. In contrast, D-MNC still shows obvious pre-edge peaks, whose intensities gradually decrease by ~2.46/~6.86% (Fig. 3c and Supplementary Fig. 11)[27–29]. Decrease in integrated intensity of the pre-edge peak until 10th and 100th cycle is dominantly correlated with activation of $Li_2MnO_3$ accompanying oxygen loss and overall structural degradation, respectively. TM-O hybridization and oxygen evolving due to surface reactions are very relevant. By comparing the Bet surface area of both pristine D-MNC and O-MNC, D-MNC (5.08 m² g⁻¹) has a higher area than O-MNC (3.36 m² g⁻¹). This implies that

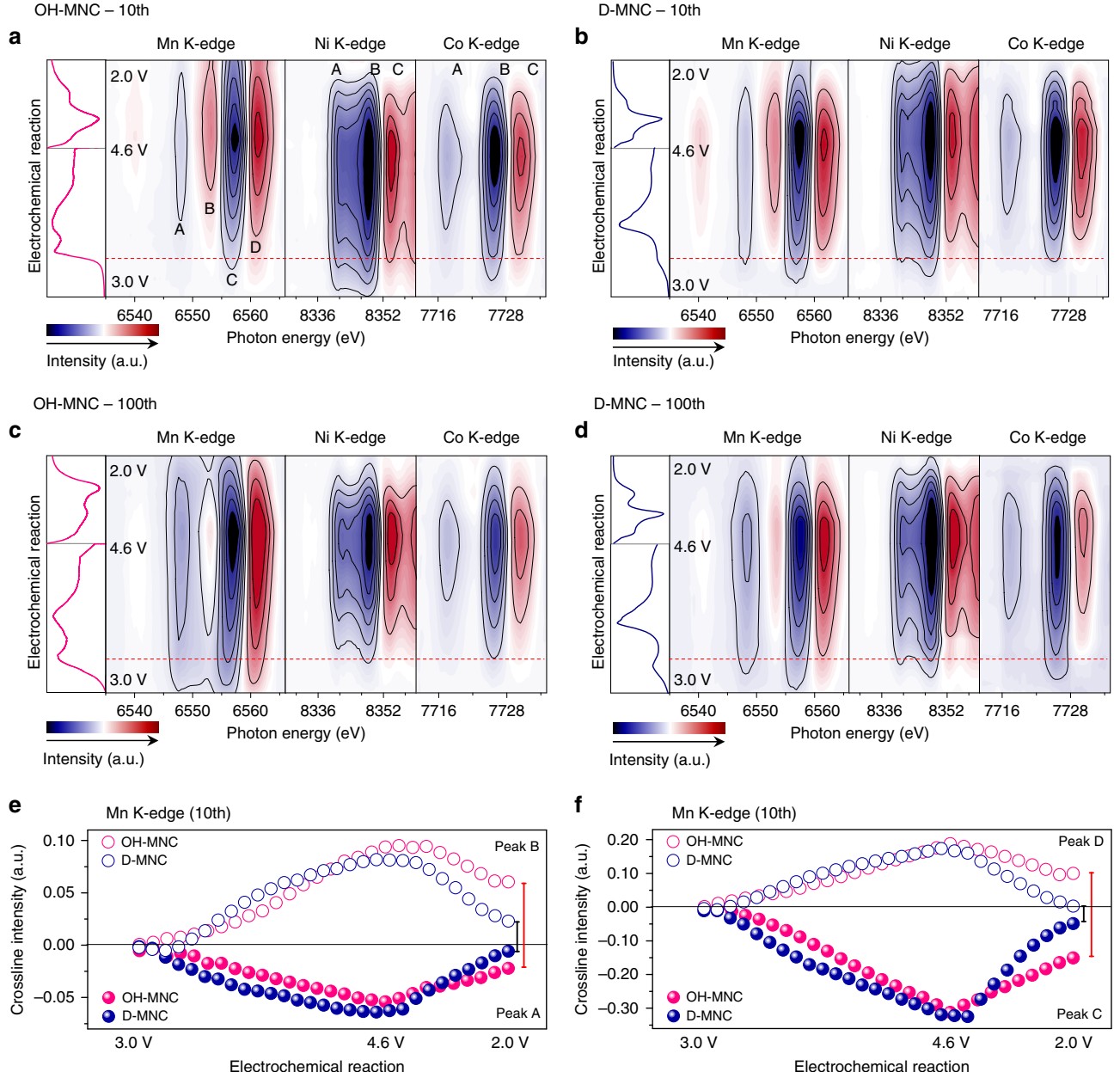

**Fig. 2** Transition-metal K-edge *operando* XANES variation during cycling. Mn, Ni, and Co K-edge *operando* XANES spectra (2D contour plot) and voltage profiles of **a** OH-MNC 10, **b** D-MNC 10th, **c** OH-MNC 100th, and **d** D-MNC 100th; Crossline intensity of Mn K-edge *operando* XANES spectra peaks **e** A and B; **f** C and D during 10th cycle; Electrodes were cycled at 0.5C-rate charge 1.0C-rate discharge condition

the more surface reaction proceeds compared to O-MNC when the reactivity of the materials is assumed to be the same. However, structural stability and O K-edge change in D-MNC were less and more stable compared to OH-MNC. This suggests that the hybridization feature of TM-O contributes a lot to surface reactivity as well as structural stability. Therefore, surface reaction is highly relevant with TM-O hybridization.

On oxidation (lithium extraction) of OH-MNC and D-MNC, electrons are removed from the TM orbital and electron transfer from the O2p band to the TM band occurs to compensate charge neutrality with hole creation in the O2p band[33,34]. Therefore, repeated charge-discharge process with high voltage condition leads to excessive hole creation in the O2p orbital and labile oxygen state. Consequently, the conservation of pre-edge peak intensities in D-MNC after 10th and 100th cycle means that the

covalent bonding character of O-3dTM loses less oxygen from the lattice during cycling even at a high voltage, which originated from more covalent and high possibility of hole-delocalization on $Mn^{3.53+}$–O bonding in contrast to $Mn^{3.65+}$–O bonding of OH-MNC (Fig. 3d).

**Microscopy atomic arrangement analysis.** A combination of high-angle annular dark-field scanning transmission electron microscopy (HAADF-STEM) and energy dispersive X-ray spectroscopy (EDS) reveals the correlation between oxygen stability and structural arrangement. Delmas et al, successfully investigate the structural rearrangement of 1st cycled Li-excess 3dTM material along $[100]_{mono}$ zone axis[36]. In order to analyze the structural characteristic of Li-excess material correctly, Atomic arrangement along $[100]_{mono}$ direction where we can observe

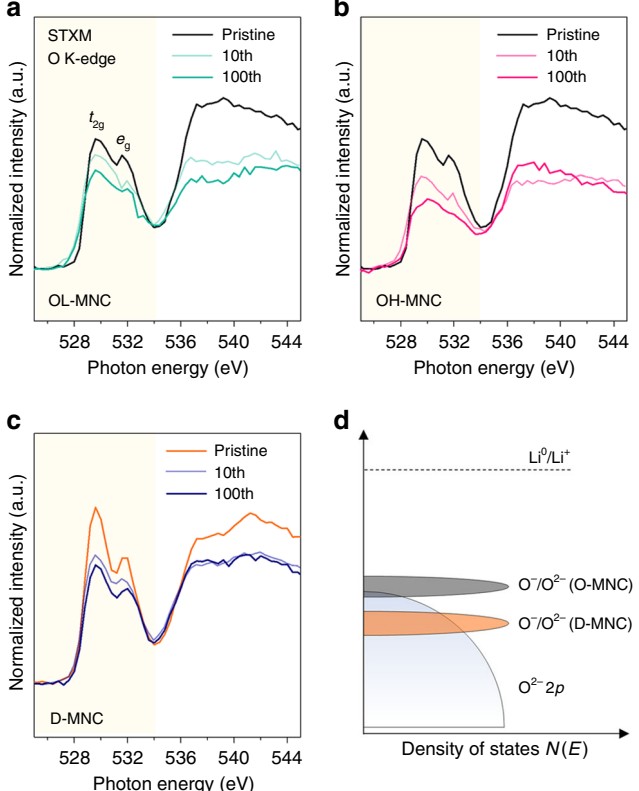

**Fig. 3** Oxygen stability of TM-O bonding and electron hole localization on oxygen. Oxygen K-edge SXAS spectrum of **a** OL-MNC **b** OH-MNC **c** D-MNC; the spectra collected on Pristine, 10th cycled and 100th cycled (discharged state) particle. **d** Schematic diagram of the possible $O^-/O^{2-}$ redox couple position based on TM-O covalency

simulated HAADF/ABF-STEM image of $I41$ structure model along both $[310]_{mono}$ and $[100]_{mono}$ directions.

Interestingly, the cation-disordered structure of D-MNC (discharged state) is preserved without further phase transition even after 100 cycles compared to OH-MNC (Fig. 4c, d and Supplementary Fig. 15). Furthermore, EDS analysis reveals the oxygen deficiency according to the distance from the outer surface (Supplementary Fig. 16, 17 and Supplementary Table 6). O/TM ratio below 1.33 indicating $TM_3O_4$ phase was detected at a distance of ~23 nm from the surface of OH-MNC and within ~5 nm for D-MNC, consistent with the results of STXM. Continuous oxygen loss and consequent cation migration ($O_{TM}/O_{LiTM} \rightarrow O_{Li} \rightarrow$ tetrahedral site) resulting in a three-step phase transition (Layered $\rightarrow$ $LiTM_3O_4$ phase $\rightarrow$ $TM_3O_4$ phase) in both $[310]_{mono}$ and $[100]_{mono}$ directions (Supplementary Fig. 18 and Supplementary Note 2).

**Atomic-selective structural analysis**. In order to reveal the relationship between atomic rearrangement and TM redox mechanism changes on prolonged cycling, XRD profiles and extended X-ray absorption fine structure (EXAFS) spectra were investigated. The XRD patterns of OL-MNC, OH-MNC, and D-MNC show significant peak shifts towards lower angles and the super-lattice peak disappears from their XRD profiles after 10 cycles. However, even if severe voltage decay is occurred in OH-MNC during 100 cycles, a slight peak shift towards lower angles is observed (Supplementary Fig. 19). Although the XRD profiles reveal the expansion of lattice, they do not indicate mutual atomic arrangements and local atomic transitions around the specific TM related to voltage decay.

The radial distribution function (RDF) of the Mn, Ni, and Co K-edge $k^3\chi(k)$ EXAFS spectra providing local structure of a specific TM element was investigated and the interatomic distance was quantitatively analysed. Additionally, correlations between spectra peaks and structural information are schematically described in Supplementary Fig. 20. Interestingly, the lower TM-A/C peak and higher TM-B peak were observed in pristine D-MNC than those of pristine O-MNC. This indicates that the pristine D-MNC structure has a relatively disordered arrangement in the TM and Li layer than the pristine O-MNC structure does (Supplementary Fig. 21). After the 100th cycle, a significant decrease in Mn–O and Ni/Co-TM peaks were observed in 100th-cycled OH-MNC (discharged state) compared with the OL-MNC (Fig. 5a and Supplementary Fig. 22), which originated from irreversible Ni/Co ion migration (intra-layer to $O_{Li}$ in the inter-layer) and evolution of layered $MnO_2$-type structure caused by lithium extraction and oxygen loss from $Li_2MnO_3$ and $LiTMO_2$ phases[11,42–45]. In contrast, overall peak features in all RDFs for D-MNC (discharged state) were relatively constant, compared with OH-MNC, indicating the relative preservation of TM local structure surrounded by oxygen and TM even during 100th cycle.

In order to investigate real-time structural variation and consequent atomic rearrangement after 10th and 100th cycle, the RDFs of TM-O and TM-A K-edge peak at all TM were plotted with a 2D contour map as a function of electrochemical reaction for clarity (Fig. 5b, c). During the charge–discharge, the 2D pattern intensities for each bonding pair in D-MNC keep a constant feature or become more intense from 10th to 100th cycle, while OH-MNC at 100th cycle distinctly shows a significant decrease in pattern intensity compared to OH-MNC at 10th cycle. The diminishing intensities in OH-MNC from 10th to 100th cycle indicate more oxygen loss from the lattice, less covalent bonding, and a static disorder of edge-shared TM coordination (ordered-to-disordered phase transition). On the other hand, the constant intensity of D-MNC for both 10th and

$Li_2MnO_3$ phase is additionally needed to increase the structural analysis accuracy. Therefore, we observed that structural deterioration from the layered to the spinel phase occurs in both directions through not only the $[310]_{mono}$ zone but also the distinguishable $[100]_{mono}$ zone. Through the EDS analysis, sequence of structural deterioration associated with oxygen deficiency were proposed. Pristine O-MNC features a well-ordered layered structure without cation disordering within the Li layer along the $[100]_{mono}$ and $[310]_{mono}$ directions. Both TM-TM-TM arrangement of $LiTMO_2$ structure and Li-TM-TM of the $Li_2TMO_3$ structure coexisted in O-MNC along the $[100]_{mono}$ direction. Additionally, a spot-streak in the FFT pattern of pristine O-MNC resulting from the coexistence of $Li_2TMO_3$ structure along $[100]_{mono}$, $[\bar{1}10]_{mono}$, and $[1\bar{1}0]_{mono}$ indicates stacking faults in O-MNC[37]. In contrast, cation disordering within the Li layer were observed in pristine D-MNC along $[310]_{mono}$ and $[100]_{mono}$ directions (Fig. 4a, b and Supplementary Fig. 12)[37–41].

After 100 cycles, OH-MNC (discharged state) shows severe phase transition along both $[310]_{mono}$ and $[100]_{mono}$ directions, originating from cation migration between octahedral sites within the TM ($LiTMO_2$ phase) / LiTM ($Li_2TMO_3$ phase) layers and Li layers (denoted as $O_{TM}$, $O_{LiTM}$, and $O_{Li}$) (Fig. 4c, d). Migration of cations from $O_{TM}$ and $O_{LiTM}$ to $O_{Li}$ causes propagation of Domain B ($LiTM_3O_4$ phase) into the internal structure during cycling. Continuous Li intercalation and oxygen loss leads to additional cation migration from octahedral site to tetrahedral site, resulting in Domain A ($TM_3O_4$ phase) at the outer surface of OH-MNC which directly contacted with electrolyte (Supplementary Fig. 13 and 14). These structures match well with the

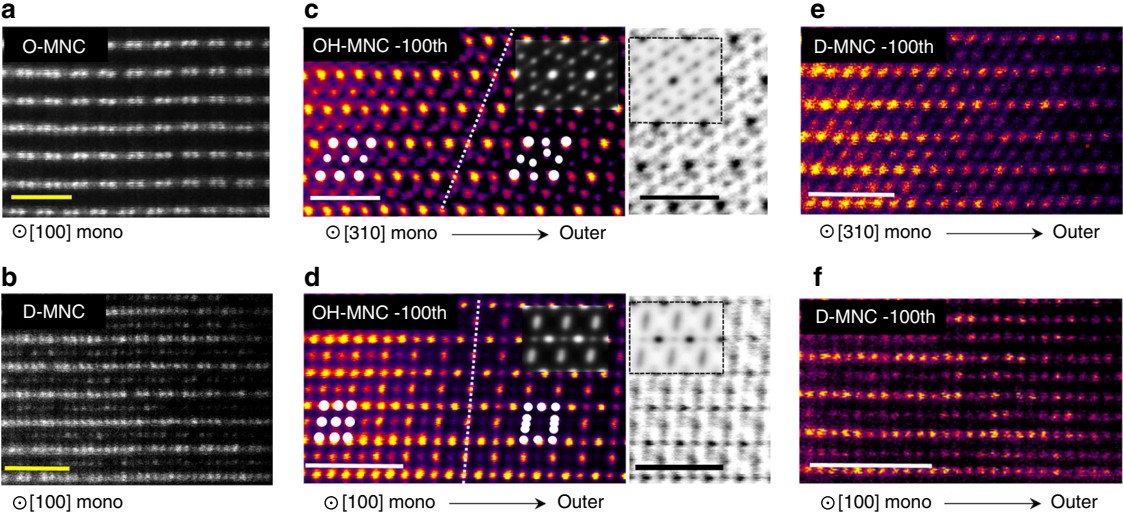

**Fig. 4** Evolution of atomic structure. HAADF-STEM image of the **a** $[100]_{mono}$ Pristine O-MNC, **b** $[100]_{mono}$ Pristine D-MNC particle. **c** $[310]_{mono}$ OH-MNC, **d** $[100]_{mono}$ OH-MNC, **e** $[310]_{mono}$ D-MNC and **f** $[100]_{mono}$ D-MNC particle (discharged state) after 100 charge-discharge cycles. Enlarged HAADF/ABF-STEM images of the surface region in images **c** and **d** with simulate HAADF/ABF-STEM images (in the inset) are presented; Scale bar denotes 1 nm

$100^{th}$ cycle reflects less oxygen loss from lattice and low possibility of cation disordering (disordered-to-disordered). Considering the RDFs, Soft-XAS, and XANES data in Fig. 2, we conclude that the stability of Mn–O bonding in Li-excessed $3d$TM layered oxides is highly correlated with the overall reversibility of TM-O and immobilization of TM ion during cycling. The change in the redox reaction of a material is related to the oxidation number of the TM that constitutes the material, and the change in the oxidation number is closely related to the structural change.

The OH-MNC, which has a relatively higher oxidation number in the 10th charged state, has fewer shared electron between TM and O resulting in a weak covalent TM-O bond. Therefore, the higher the oxidation number in the charged state, the more unstable TM-O bonding will result in a structurally unstable state. Covalent bond of Mn–O bonding which is the main structure of the Li-excess $3d$TM oxide materials, is weakened due to the excessive delithiation/charge process for high capacity expression resulting in the labile oxygen release with oxygen vacancy formation in structure resulting in hole generation during the 10th cycle. Electrons originating from oxygen vacancy ($O_O = V''_O + 2e^- + 0.5O_2$) occupy the $TM3d t_{2g}$ states resulting in reversible hole decrease of Mn ion. As a result, the Mn ions with hole, which can reversibly receive the electrons, are more in D-MNC by exhibiting reversible characteristics which originated from gradual increase (decrease) in hole (TM-O covalency) during 10th cycle compared to OH-MNC. When undergoing a long-term charge/discharge process, continuous unstable Mn–O bond formation and existence of oxygen vacancies cause unstable structure sequentially. Ni and Co ions were constantly migrating to stabilize the structural instability, and the disordered phase formed by cation migration results in abnormal behaviour.

Therefore, maintenance of TM-O bonding and immobilization of TM ion in D-MNC, especially Mn–O bonding and Ni/Co ion, is expected to highly affect the stable TM redox activity and the suppression of voltage decay during cycling, unlike in OH-MNC with ordered structure.

### How atomic arrangement affect structure stability of Li-excess material.

Based on the multilateral analysis results, we described oxygen-centred macroscopic structure models of OH-MNC and D-MNC with three types of de-lithiated $M_6O$ octahedron ($M_4O$) to explain the fundamental reason for the difference in oxygen stability and structural reversibility according to atomic arrangement (Fig. 6a). Each $O^{2-}$ ion is coordinated by four cations and two lithium vacancies ($V_{Li}$) with different portion of $Li^+$ and transition-metal in TM and Li layer. 2TM-O-2Li $[2V_{Li}]$, 3TM-O-1Li $[2V_{Li}]$ and 4TM-O-0Li $[2V_{Li}]$ models represent the base structures of de-lithiated $C2/m$, $R\bar{3}m$, and cation-disordered $R\bar{3}m$, respectively (Fig. 6a). High number of TM around oxygen increases the $O2p$-$TM3d t_{2g}$ orbital hybridization and decreases the possibility of holes localization on the oxygen sites at charged states compare to high number of Li around oxygen[28,29].

Microscopic structural analysis along $[310]_{mono}$ and $[100]_{mono}$ directions and EXAFS spectra revealed that O-MNC with a well-ordered structure has long-range ordering of 2TM-O-4Li octahedron. In contrast, 2TM-O-4Li ordering is broken in D-MNC because the disordered cations in the Li layer form 3TM-O-3Li and 4TM-O-2Li octahedrons in the $[100]_{mono}$ structure (Supplementary Fig. 23). At de-lithiated state, the presence of 4TM-O-0Li $[2V_{Li}]$ octahedrons in structure increases the number of stable oxygen and possibility of hole delocalization on oxygen related with high covalency of TM-O bonding and low oxidation number in D-MNC. In perspective of cation, stable oxygen resulting from short-range ordered 2TM-O-2Li $[2V_{Li}]$ octahedron is coordinated around the TM of D-MNC unlike in OH-MNC (Fig. 6b). High-voltage condition causes excessive oxidation of TM in the structure, resulting in more oxygen loss on TM-O bonding, especially $Mn^{3.65+}$–O bonding (less covalent) compared to $Ni^{3.08+}$/$Co^{2.98+}$–O bonding. When charge compensation is continuously repeated, series of oxygen loss occurs from overall TM-O bonding, and Ni/Co cation migration is preferentially occurred in the ordered structure of OH-MNC than in the cation-disordered structure of D-MNC (Fig. 6c).

### Discussion

Structural arrangement with de-lithiated oxygen-centred octahedron ($M_4O$) and interactions between those octahedrons are important reasons for Li-excess material degradation. By combining spectroscopy and microscopy techniques, we reveal that the cation distribution in Li and TM layers determines the ordering of octahedron ($M_6O$) and different TM-O bonding covalency. At charged state, numbers of labile oxygen (high possibility of hole localization) arisen from excess oxidation of

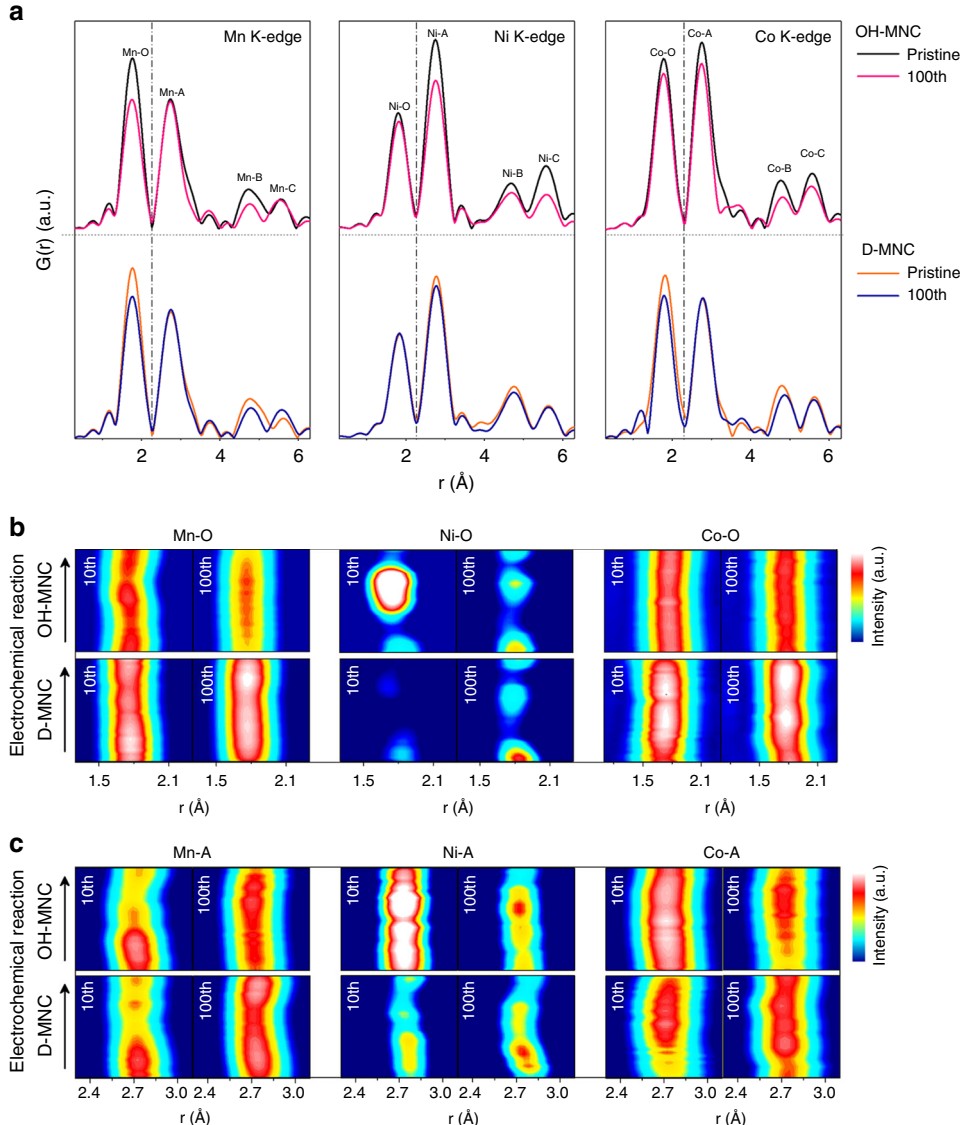

**Fig. 5** Migration of specific elements due to the structural instability. **a** Radial distribution function (RDF) of Mn, Ni, and Co K-edges $k^3$-weighted EXAFS spectra for pristine and 100th cycled (discharged state) electrode of OH-MNC and D-MNC; Comparison of OH-MNC and D-MNC K-edge *operando* EXAFS spectra (2D contour plot) collected during 10th and 100th cycle. Spectra represents **b** TM-O peak and **c** TM-A peak variation of Mn, Ni, and Co K-edges

well-ordered $Li_2TMO_3$ phase leads to decrease of TM-O bonding covalency, especially Mn–O bonding, resulting in oxygen loss from lattice and irreversibility of the overall TM-O bonding. In particularly, Ni/Co ions surrounded with labile oxygens and oxygen vacancies significantly migrate into Li vacancies, and consequently causes 'layered to $TM_3O_4$' phase transition during the cycling. These features finally bring about the inactive-redox and active-redox reaction in Ni/Co and Mn, respectively, causing the voltage decay phenomenon. By comparing ordered and cation-disordered Li-excess 3dTM material, we advance the basic understanding of voltage decay phenomenon, in terms of de-lithiated oxygen-centred octahedron ($M_4O$), and propose candidate structure model to suppress voltage decay, having partially distributed cation-disordered site, not a well-ordered structure.

## Methods

**Sample preparation**. $Li_{1.15}Mn_{0.51}Co_{0.17}Ni_{0.17}O_2$ (denoted as O-MNC) and $Li_{1.09}Mn_{0.55}Ni_{0.32}Co_{0.043}O_2$ (denoted as D-MNC), were prepared by co-precipitation method. Stoichiometric amounts of $MnSO_4·5H_2O$ (99.0 %, JUNSEI), $NiSO_4·6H_2O$ (98.5~102%, SAMCHUN) and $CoSO_4·7H_2O$ (98.0 %, SAMCHUN) were used as the starting materials. An aqueous solution of the reagents at a

concentration of 3.0 mol $L^{-1}$ was pumped into a continuously stirred tank reactor (CSTR; 4 L) under a nitrogen atmosphere. The pH was adjusted to 10.5 with a 3.0 mol $L^{-1}$ solution of NaOH, and a desired amount of a solution of $NH_4OH$ (4 mol $L^{-1}$) as a chelating agent was also separately fed into the reactor. The obtained solid was filtered, washed many times with distilled water, and then dried at 110 °C for 12 h. The dried powder was thoroughly mixed with $LiOH·H_2O$. (98.0%, Sigma-Aldrich) and calcined at 900 °C for 10 h. To analyze the material characteristics, pristines and electrodes recovered in the discharged state at ~2.00 V after 10 and 100 cycles, respectively,

**Transmission electron microscopy (TEM) analysis**. Structural characterization of the samples was carried out using SEM (Verios 460, FEI). HR-TEM (JEM-2100F, FEI) was conducted for detailed analysis. EDS was utilized in HR-TEM (EDS, Aztec, Oxford). XRD (D/Max2000, Rigaku) was carried out for the powder analysis using Cu-Kα radiation, a scan range of 10–90°, a step size of 0.005°, and a counting time of 5 min. Rietveld profile refinements were performed using the GSAS suite of programs. HAADF-STEM images were obtained with aberration-corrected JEM-2100F electron microscopes operated at 160 kV using a convergence semi-angle of 22 mrad. The HAADF inner and outer collection semi-angle was 54 mrad and 220 mrad, respectively. Simulated HAADF/ABF-STEM image and FFT pattern were obtained by using Dr.probe[46] and CrysTbox[47] program.

***Operando* X-ray absorption spectroscopy (XAS)**. *Operando* transition metal (TM = Mn, Ni, Co) K-edge X-ray absorption fine structure (XAFS), X-ray

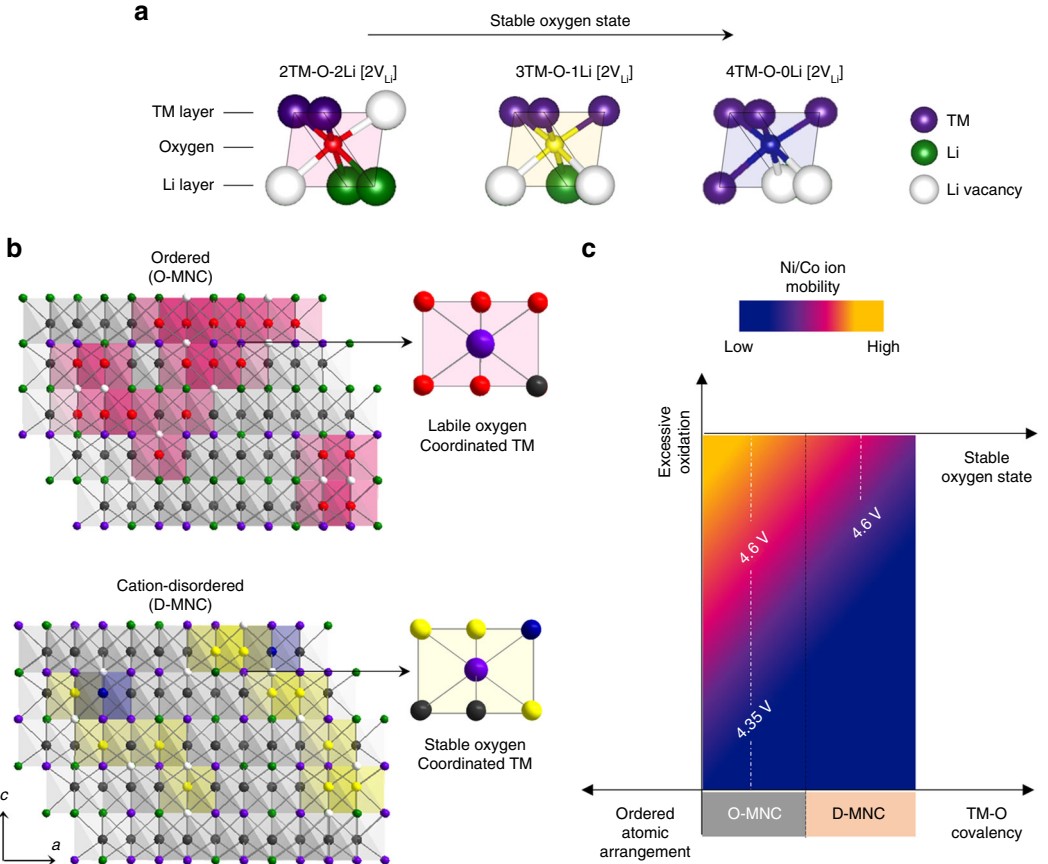

**Fig. 6** Factors to consider in interpreting the structural stability of high capacity cathode material. **a** Three de-lithiated oxygen-centred octahedron ($M_4O$) of Li-excess material. Each oxygen anions coordinated by four cations and two lithium vacancies ($V_{Li}$) with different portion of $Li^+$ and transition-metal in TM and Li layer. 2TM-O-2Li [$2V_{Li}$], 3TM-O-1Li [$2V_{Li}$] and 4TM-O-0Li [$2V_{Li}$] octahedron represent the base structure of de-lithiated $Li_2TMO_3$, $LiTMO_2$ and cation-disordered-$LiTMO_2$, and the degree of stability on centred oxygen represented as different color (Red is highly labile). **b** Oxygen-centred macroscopic structure model of de-lithiated state OH-MNC and D-MNC along [100]$_{mono}$. **c** Schematic of structure stability according to the material characteristics and electrochemical conditions

absorption near edge structure (XANES) and extended X-ray absorption fine structure (EXAFS), were collected on the BL10C beam line (WEXAFS) at the Pohang light source (PLS-II) with top-up mode operation under a ring current of 360 mA at 3.0 GeV. From the high-intensity X-ray photons of the multipole wiggler source, monochromatic X-ray beams could be obtained using a liquid-nitrogen-cooled double-crystal monochromator (Bruker ASC) with available in situ exchange in vacuum between a Si(111) and Si(311) crystal pair. The Si(111) crystal pair was used for TM K-edge XAFS measurements. Real-time TM K-edge X-ray absorption spectroscopic data during discharging and charging processes were recorded for 10th-cycled OL-MNC, 10th/100th-cycled OH-MNC, and 10th/100th-cycled D-MNC electrodes assembled in a home-made in situ electrochemical cell with polyimide film windows (Swagelok-type cell), in transmittance mode using $N_2$ gas-filled ionization chambers (IC-SPEC, FMB Oxford) for the incident and transmitted X-ray photons. Higher-order harmonic contaminations were eliminated by detuning to reduce the incident X-ray intensity by ~30%. Energy calibration was simultaneously carried out for each measurement with reference TM foils placed in front of the third ion chamber. All XAFS data were measured using on-the-fly mode with scanning time of 1 min for one spectrum per every 5 min. Under the 0.5 C-rate charging and 1C-rate discharging, XAFS spectra above 20 scans in charging and above 10 scans in discharging have been acquired independently for all electrodes. The data reductions of the experimental spectra to normalized XANES and Fourier-transformed radial distribution functions (RDFs) were performed through the standard XAFS procedure. Using AUTOBK and FEFFIT modules in UWXAFS package, the $k^3$-weighted TM K-edge EXAFS spectra, $k^3\chi(k)$, have been obtained through background removal and normalization processes on the edge jump. In order to present effective radial distribution functions (RDF) for each sample during charge-discharge, the $k^3\chi(k)$ spectra have been Fourier-transformed (FT) in the $k$ range between 2.5 and 12.0 $Å^{-1}$.

**Scanning transmission X-ray microscopy (STXM).** STXM images and Soft X-ray absorption spectroscopy (SXAS) data were obtained at the Pohang Light Source

(10A beam line) using the monochromatic soft X-rays of the synchrotron source. To obtain XAS spectra at the O K-edge, stacks of STXM images were obtained by changing the incident photon energy, while keeping the focal position at the sample plane, and then spectroscopic data from a specific point or area (in the lateral plane) were collected from the stack images through the aXis2000 software package. A monochromator resolution of 0.1 eV was used for recording SXAS spectra. Image stacks were acquired at 520–560 eV (O K-edge) to extract the X-ray absorption spectra from the surface region of samples.

**Electrochemical characterization.** For fabrication of the cathode electrode, the prepared powders were mixed with carbon black and polyvinylidene fluoride (8:1:1) in N-methylpyrrolidinon. The obtained slurry was casted onto Al foil with active material loaded at 4.5–5.0 mg cm$^{-2}$ and roll-pressed. The electrodes were dried overnight at 110 °C in dry oven. The electrochemical performances of the prepared O-MNC and D-MNC were measured in tests using 2032 coin-type cells. The coin cells were assembled with Li metal as the counter and reference electrodes. In order to assemble the cell, CR2032 (half-cell) cell were utilized in argon-filled glove box. The electrolyte was a solution of 1.3 M LiPF$_6$ in ethylene carbonate (EC)/ethylmethyl carbonate (EMC)/ diethyl carbonate (DEC)/(2/5/3, by volume) with 5.0% of fluoroethylene carbonate, 0.5% of vinylene carbonate and 1.0% of tris (trimethylsilyl) phosphite. As a separator, microporous polyethylene (Celgard) was used. The galvanostatic charge-discharge cycling was carried out between 2.0 and 4.6 V (vs. Li/Li$^+$) at 25 °C temperature. To obtain accurate data, 4 cells per sample were tested.

**Data availability**. The authors declare that the data supporting the findings of this study are available within the article and its Supplementary Information Files. All other relevant data supporting the findings of this study are available on request.

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

## Acknowledgements

This work was supported by IT R&D program of MOTIE/KEIT (Development of Li-rich Cathode and Carbon-free Anode Materials for High Capacity/High Rate Lithium Secondary Batteries, 10046309). Also, Research Funds (1.180019.01 and 1.180033.01) and of Ulsan National Institute of Science and Technology (UNIST) was greatly acknowledged.

## Author contributions

S.M. conceived and designed the experiments. S.M. performed and analyzed the major part of the experiments and the results; W.C., W.J. and J-G.H. assisted with sample preparation and electrochemical test; M.Y assisted with XRD analysis. G.N and H.J assisted with XAS measurement. J.H and N-S.C offered electrolyte; Y.Y. assisted with

TEM analysis. M.G.K assisted with XAS analysis. S.M. wrote the paper. W.C., M.G.K., and J.C., discussed the results and revised or commented the manuscript.

## Additional information

**Competing interests:** The authors declare no competing interests.

