## [Peer Review File · Nature Communications]

Reviewers' comments:

Reviewer #1 :

The authors reveal the relationship between cation arrangement and voltage decay of lithium rich cathode materials by comparing the electrochemical behaviors and the associated structural changes of two samples, ordered and pre-disordered materials. The results and conclusions are reasonable, however, some critical issues need to be addressed before the paper can be considered for publication.

1. I don't quite understand the definition of "pre-disordered" phase. The XRD patterns of two samples seem similar. The authors would give some explanation on that.
2. The refinements of XRD patterns only would be insufficient to find out the cation order/disorder in lithium rich samples. Neutron diffraction patterns may be helpful. The characterization of the degree of cation disorder in samples will be important for this work.
3. In Figure 3, the authors show O K-edge XAS curves of samples after 10 and 100 cycles. Could the authors explain the changes of intensity observed in the high energy range (>536 eV)?
4. In Figure 4, the authors show STEM images of samples after 100 cycles to indicate the cation migration/disorder. Delmas et al., and Dahn et al., proposed the core-shell model in cycled lithium rich electrode sample. The authors should consider this in the discussion.
5. The authors study the voltage decay process of lithium rich sample upon cycling, but only focus on the discharge process. How about the charge process?

Reviewer #2 :

This manuscript is a comparative study of the redox reactions and cation migrations of the ordered and pre-disordered Li-excess cathodes. The key claim of this work is the contrast on voltage fade of the two systems corresponds to the different oxygen redox activities and cation migrations. While such discussions could be found in previous literature, I think providing direct comparisons of two systems still delivers important information and clarification on this important topic. Additionally, the final discussion provides a new way to look at the local molecular configurations, which is intriguing and interesting to a broad audience. I therefore think this work deserves the publication in NC. In the meantime, I think the authors should clarify the following points to improve the clarity and impact of the work.

1. To me, the two comparative systems of the ordered and pre-ordered Li-excess materials are wise choices in this study. However, I do not find clear message on the material contrasts and definitions of the "ordered" and "pre-disordered". It seems the different materials are synthesized based on only different stoichiometry with all other procedures the same (Supplementary)? How to define the so-called "ordered" and "pre-disordered" while one could see clearly the Li-TM ordering peak in XRD for both systems?
2. Page 5, paragraph 1, how does the relatively more reversible Mn redox is associated with "relatively covalent bond with surrounded oxygen compare to that of OH-MNC"? Please note covalency of a battery electrode system changes significantly upon the charging state, it is hard to understand what the authors try to describe here on the two materials in general through cycling. Confusing but intriguing discussions could be found at several other places too, e.g., page 6, 1st paragraph, what does it mean by saying "structural instability associated with irreversible Mn ion redox ... causing the abnormal behavior ... of Ni/Co and Mn..."? Structural instability is further discussed later in the paper

through EXAFS, but I would appreciate if the authors could clearly write their opinions and conclusions in such kind of important discussions.

3. While I think the authors did a nice job on the cationic redox and RDF analysis, the O-K STXM study is poorly presented, making it hard to understand or accept the final schematic of Fig.3d.

3a. First, as the most important technical parameter, where is the probe location of the STXM for the data shown? It was clearly shown recently (Nat Comm 8, 2091 2017) that STXM close to the surface of the particle gives very different signal on O-K, compared with STXM collected towards the center of the particle (more bulk signal). Only the STXM signal around the central area of the particle could detect the intrinsic oxygen redox signal.

3b. Another critical but missing information: typical oxygen redox discussions based on O-K are based on the spectral contrast between the charged and discharged states, as in the several references already cited in this manuscript. But here, this critical information is also missing, i.e., what is the charge/discharge state of the "cycled" particle here?

3c. I would like to say that the discussion of the pre-edge of O-K based on TM-O hybridization strength (covalency) here is a much better job than the serious misinterpretation of the O-K pre-edge in many high-profile papers, including some Refs here. It has been noted by more and more researchers now that such pre-edge features do not represent the intrinsic oxygen redox behavior, e.g., J Power Sources 389, 188 2018. Therefore, I would agree with the general claim from the authors based on the overall better defined peaks in d-MNC. But to me, such contrast is really because less oxygen evolving out from d-MNC leads to less surface reaction, leaving better defined TM-O hybridization features. While this comment does not affect the central claims, I just do not see how one could get the schematic of Fig. 3d "based on TM-O Covalency variation observed in O K-edge SXAS".

4. Also, please define the "cycled" samples for Fig. 4 and Fig. 5 too. Are they charged or discharged samples after cycling?

5. There have been detailed studies based on HAADF-STEM of 3d TM based Li-rich compounds, e.g., JPCC 119, 75 (2015) and some others. In addition to the Ru system (Ref. 26), these works also discuss the cation migration into octahedron and tetrahedron during cycling and its reversibility. I would like to ask whether the microscopic observations here are generally consistent with these previous works or not. In whatever case, I think Fig. 6 is an interesting angle to look at the local configurations.

Reviewer #1 :

The authors reveal the relationship between cation arrangement and voltage decay of lithium rich cathode materials by comparing the electrochemical behaviors and the associated structural changes of two samples, ordered and pre-disordered materials. The results and conclusions are reasonable, however, some critical issues need to be addressed before the paper can be considered for publication.

1. I don't quite understand the definition of "pre-disordered" phase. The XRD patterns of two samples seem similar. The authors would give some explanation on that.

Response) Thank you for your question and sorry for confusing definition. We conducted additional detailed XRD Rietveld refinement to provide the overall structure disordering information of both materials (revised Supplementary Table S1) and we revised XRD patterns with magnification to show the broad Full Width at Half Maximum (FWHM) well indicating the low crystallinity (atomic disordering) of materials (revised Supplementary Figure S2). Furthermore, we redefine the both materials in the revised manuscript (page3) 'Li_{1.15}Mn_{0.51}Co_{0.17}Ni_{0.17}O₂ composition with well-ordered layered phase and long-range ordered Li-TM-TM arrangement (denoted as O-MNC) and Li_{1.09}Mn_{0.55}Ni_{0.32}Co_{0.043}O₂ composition with cation- disordered layered phase and short-range ordered Li-TM-TM arrangement (denoted as D-MNC)' based on the XRD patterns, Rietveld refinement, High-Angle Annular Dark-Field Scanning (HAADF)-STEM image (Both pristine O-MNC and D-MNC in revised Supplementary Figure S12), Extended Absorption Fine Structure (EXAFS) spectra (Both pristine O-MNC and D-MNC in revised Supplementary Figure S21) and Main Figure. 6 structure model. Although the D-MNC does not have a perfect disordering structure (Rock salt phase) breaking all of Li-TM-TM ordering resulting in big-change of XRD patterns, it can be seen that cations of D-MNC in TM layer are relatively disordered in Li layer with retains the basic structure of Li-excess 3d-transition-metal layered oxide material, which confirmed from XRD Rietveld refinement result, HAADF-STEM and TM K-edge EXAFS spectra of both pristine O-MNC and D-MNC.

2. The refinements of XRD patterns only would be insufficient to find out the cation order/disorder in lithium rich samples. Neutron diffraction patterns may be helpful. The characterization of the degree of cation disorder in samples will be important for this work.

Response) I fully agree with reviewer comment. There is a lack of discussion on degree of cation disorder by using XRD alone. Therefore, we insert additional XRD Rietveld refinement result showing cation disorder in the structure. Furthermore, we performed atomic-scale structural analysis, which provides an image of atomic arrangement and local structure information such as bonding distance, coupling distance disorder, the number and type of atoms located around through scanning transmission electron microscopy (STEM) and synchrotron x-ray absorption spectroscopy (XAS) of atomic scale, especially, High-Angle Annular Dark-Field Scanning (HAADF) STEM image and Extended Absorption Fine Structure (EXAFS). As a result, it was possible to derive the results related to the ordering of cation in accordance with the XRD data in the pristine D-MNC and O-MNC materials, and these sufficient result and analysis indicating the cation disordering in Li-excess 3dTM material are originally mentioned at the beginning of the 'Microscopy atomic arrangement analysis' part (revised manuscript page8 and Supplementary Figure S12) and 'Atomic-selective structural

analysis' part (revised manuscript page9 and Supplementary Figure S21). Therefore, we would appreciate if you would consider the results.

3. In Figure 3, the authors show O K-edge XAS curves of samples after 10 and 100 cycles. Could the authors explain the changes of intensity observed in the high energy range (>536 eV)?

Response) The region over 536 eV indicates the transition-metal 4sp band mixed with O2p state. It is hard to find the meaning of intensity change during cycle, However, It is possible to observe the change in the hole through the region ratio of 3d/4sp.¹ Therefore, we revised and inserted additional figure in the revised SI (Supplementary Figure S11) with explanation.

'After 10th cycle, we expected that the activation process of Li₂TMO₃ phase with oxygen evolutions (oxygens with shared electrons) causes the increase of hole (increase ratio of 3d/4sp). However, OH-MNC shows significant variation and low number of holes compared to D-MNC. After 100th cycle, the variation of ratio is proportion to the voltage decay rate of OL-MNC, OH-MNC and D-MNC shown in Figure.1. From the results, we can expect that significant increase and low number of holes after 10th cycle causes irreversible behavior of redox reaction. Furthermore, Cation migration and atomic rearrangement with decreasing the hole (decrease ratio of 3d/4sp) occurs to decrease the instability of structure originated from continuous oxygen evolution during cycling'. These sentence was added in the revised SI (page19).

1. De Groot, F. M. F.; Grioni, M.; Fuggle, J. C.; Ghijsen, J.; Sawatzky, G. A.; Petersen, H., Oxygen 1s X-ray-absorption edges of transition-metal oxides. *Physical Review B* **1989**, 40 (8), 5715-5723.

4. In Figure 4, the authors show STEM images of samples after 100 cycles to indicate the cation migration/disorder. Delmas et al., and Dahn et al., proposed the core-shell model in cycled lithium rich electrode sample. The authors should consider this in the discussion.

Response) Thank you for your advice. We mentioned their successive previous studies observing atomic rearrangement after 1st cycle process in the revised manuscript (page8).

5. The authors study the voltage decay process of lithium rich sample upon cycling, but only focus on the discharge process. How about the charge process?

Response) Thank you for your additional request on manuscript content. As shown in Figure. 1, the change of the voltage profile and oxidation peak was observed during the charge process. We performed the in-situ XANES and in-situ EXAFS analysis to show the change in the both charge and discharge process during the 10th and 100th cycle, and the results are represented in main Figure. 2 and Figure. 5, respectively. Particularly, in Figure. 2, 'the overall charge/discharge redox reaction change during the cycle test was observed. In the 100th charge process of OH-MNC, the suppressed oxidation reaction of Ni and Co starts at higher voltage and the active oxidation reaction of Mn occurs at lower voltage compared with D-MNC in the 100th charging process, which are well matched with the tendency of dQ/dV plot.' Above additional explanations are mentioned in the revised manuscript (page6).

Reviewer #2 :

This manuscript is a comparative study of the redox reactions and cation migrations of the ordered and pre-disordered Li-excess cathodes. The key claim of this work is the contrast on voltage fade of the two systems corresponds to the different oxygen redox activities and cation migrations. While such discussions could be found in previous literature, I think providing direct comparisons of two systems still delivers important information and clarification on this important topic. Additionally, the final discussion provides a new way to look at the local molecular configurations, which is intriguing and interesting to a broad audience. I therefore think this work deserves the publication in NC. In the meantime, I think the authors should clarify the following points to improve the clarity and impact of the work.

1. To me, the two comparative systems of the ordered and pre-ordered Li-excess materials are wise choices in this study. However, I do not find clear message on the material contrasts and definitions of the “ordered” and “pre-disordered”. It seems the different materials are synthesized based on only different stoichiometry with all other procedures the same (Supplementary)? How to define the so-called “ordered” and “pre-disordered” while one could see clearly the Li-TM ordering peak in XRD for both systems?

Response) Thank you for your question and sorry for confusing definition. We conducted additional detailed XRD Rietveld refinement to provide the overall structural disordering information of both materials (revised Supplementary Table S1) and we revised XRD patterns with magnification to show the broad Full Width at Half Maximum (FWHM) well indicating the low crystallinity (atomic disordering) of materials (revised Supplementary Figure S2). Furthermore, we redefine the both materials in the revised manuscript (page3) ‘Li_{1.15}Mn_{0.51}Co_{0.17}Ni_{0.17}O₂ composition with well-ordered layered phase and long-range ordered Li-TM-TM arrangement (denoted as O-MNC) and Li_{1.09}Mn_{0.55}Ni_{0.32}Co_{0.043}O₂ composition with cation- disordered layered phase and short-range ordered Li-TM-TM arrangement (denoted as D-MNC)’ based on the XRD patterns, Rietveld refinement, High-Angle Annular Dark-Field Scanning (HAADF)-STEM image (Both pristine O-MNC and D-MNC in revised Supplementary Figure S12), Extended Absorption Fine Structure (EXAFS) spectra (Both pristine O-MNC and D-MNC in revised Supplementary Figure S21) and Main Figure. 6 structure model. Although the D-MNC does not have a perfect disordering structure (Rock salt phase) breaking all of Li-TM-TM ordering resulting in big-change of XRD patterns, it can be seen that cations of D-MNC in TM layer are relatively disordered in Li layer with retains the basic structure of Li-excess 3d-transition-metal layered oxide material, which confirmed from XRD Rietveld refinement result, HAADF-STEM and TM K-edge EXAFS spectra of both pristine O-MNC and D-MNC.

In addition, the different stoichiometry of Ni and Co ratio affect the structure of material due to the different octahedral site stabilization energy of TM ion (OSSE- Ni³⁺ : -12.67Dq / Co³⁺ : -21.33Dq).¹ A larger OSSE value for the Co³⁺ ions makes the TM migration difficult. Therefore, D-MNC with High Ni content and Low Co content shows more disorder structure compared to O-MNC with high Co content. This sentence was added in the revised SI (page8-9).

1. Choi, S.; Manthiram, A., Factors influencing the layered to spinel-like phase transition in layered oxide cathodes. *J. Electrochem. Soc.* **2002**, *149* (9), A1157-A1163.

2. Page 5, paragraph 1, how does the relatively more reversible Mn redox is associated with “relatively covalent bond with surrounded oxygen compare to that of OH-MNC”? Please note covalency of a battery electrode system changes significantly upon the charging state, it is hard to understand what the authors try to describe here on the two materials in general through cycling.

Confusing but intriguing discussions could be found at several other places too, e.g., page 6, 1st paragraph, what does it mean by saying “structural instability associated with irreversible Mn ion redox ... causing the abnormal behavior ... of Ni/Co and Mn...”? Structural instability is further discussed later in the paper through EXAFS, but I would appreciate if the authors could clearly write their opinions and conclusions in such kind of important discussions.

Response) Thank you for your additional request on important discussions and I have mentioned more detailed opinions in the revised manuscript (page10) to help understanding our experimental result and what we want to deliver to readers. I hope our answers to your questions are appropriate and helpful.

3. While I think the authors did a nice job on the cationic redox and RDF analysis, the O-K STXM study is poorly presented, making it hard to understand or accept the final schematic of Fig.3d.

3a. First, as the most important technical parameter, where is the probe location of the STXM for the data shown? It was clearly shown recently (Nat Comm 8, 2091 2017) that STXM close to the surface of the particle gives very different signal on O-K, compared with STXM collected towards the center of the particle (more bulk signal). Only the STXM signal around the central area of the particle could detect the intrinsic oxygen redox signal.

Response) Thank you for your advice and I have inserted additional figures representing the probe location as you told me and marked the signal area on existing STXM single particle image in the revised SI (page18)

3b. Another critical but missing information: typical oxygen redox discussions based on O-K are based on the spectral contrast between the charged and discharged states, as in the several references already cited in this manuscript. But here, this critical information is also missing, i.e., what is the charge/discharge state of the “cycled” particle here?

Response) The cycled particle for STXM analysis is discharged state particle. We mentioned the state of cycled particle in the revised manuscript (page6).

3c. I would like to say that the discussion of the pre-edge of O-K based on TM-O hybridization strength (covalency) here is a much better job than the serious misinterpretation of the O-K pre-edge in many high-profile papers, including some Refs here. It has been noted by more and more researchers now that such pre-edge features do not represent the intrinsic oxygen redox behavior, e.g., *J Power Sources* 389, 188 2018. Therefore, I would agree with the general claim from the authors based on the overall better defined peaks in d-MNC. But to me, such contrast is really

because less oxygen evolving out from d-MNC leads to less surface reaction, leaving better defined TM-O hybridization features. While this comment does not affect the central claims, I just do not see how one could get the schematic of Fig. 3d “based on TM-O Covalency variation observed in O K-edge SXAS”.

Response) Thank you for your question. Regarding this subject matter, we added following sentence in the revised manuscript (page7). ‘TM-O hybridization and oxygen evolving due to surface reactions are very relevant. By comparing the Bet surface area of both pristine D-MNC and O-MNC, D-MNC (5.0786 m²/g) has a higher area than O-MNC (3.3625 m²/g). This implies that the more surface reaction proceeds compared to O-MNC when the reactivity of the materials is assumed to be the same. However, structural stability and O K-edge change in D-MNC were less and more stable compared to OH-MNC. This suggests that the hybridization feature of TM-O contributes a lot to surface reactivity as well as structural stability.’ Therefore, surface reaction is highly relevant with TM-O hybridization.

In the case of Figure. 3d, we draw the figure that is expected by referring to the papers that have been conducted in previous studies.^{28, 30} Bruce et al, expected in this paper, O²⁻ 2p orbitals surrounded by Mn⁴⁺/Li⁺ forming a more ionic bonding will be located in the top of the valence band, compared to O²⁻ 2p orbitals surrounded by Ni⁴⁺ with covalent bond (d-p mixing). Based on these results, we demonstrated the possibility that the higher covalency in D-MNC would be located at the bottom of the valence band than the relatively lower O-MNC. We mentioned this reason of how get the schematic of Fig.3d in the revised manuscript (page8).

28. Luo K, Roberts MR, Hao R, Guerrini N, Pickup DM, Liu YS, *et al.* Charge-compensation in 3d-transition-metal-oxide intercalation cathodes through the generation of localized electron holes on oxygen. *Nat Chem* **8**, 684-691 (2016)
30. Luo K, Roberts MR, Guerrini N, Tapia-Ruiz N, Hao R, Massel F, *et al.* Anion Redox Chemistry in the Cobalt Free 3d Transition Metal Oxide Intercalation Electrode Li[Li_{0.2}Ni_{0.2}Mn_{0.6}]O₂. *J. Am. Chem. Soc.* **138**, 11211-11218 (2016)

4. Also, please define the “cycled” samples for Fig. 4 and Fig. 5 too. Are they charged or discharged samples after cycling?

Response) Thank you for your advice. We defined the state of “cycled” sample in legend of **Main Figure. 4 and 5 and revised manuscript.**

5. There have been detailed studies based on HAADF-STEM of 3d TM based Li-rich compounds, e.g., JPCC 119, 75 (2015) and some others. In addition to the Ru system (Ref. 26), these works also discuss the cation migration into octahedron and tetrahedron during cycling and its reversibility. I would like to ask whether the microscopic observations here are generally consistent with these previous works or not. In whatever case, I think Fig. 6 is an interesting angle to look at the local configurations.

Response) Thank you for your question. Regarding this subject matter, we added following sentence in the revised manuscript (page8). ‘In order to analyze the structural characteristic of Li-excess material correctly, Atomic arrangement along [100]mono direction where we can observe Li₂MnO₃ phase is additionally need to increase the structural analysis accuracy. Therefore, we observed that structural deterioration from the layered to the spinel phase occurs in both directions through not only the [310]mono zone but also the distinguishable [100]mono zone. Through the EDS

analysis, sequence of structural deterioration associated with oxygen deficiency were proposed.¹ We think our paper is sufficiently different from other papers which only observed phase transformation along [310]mono zone. We observed structural deterioration in both directions and links the association with oxygen deficiency through EDS, and reports a more detailed mechanism of structural deterioration. In addition, through the EXAFS analysis, we tried to increase the reliability of the structural change analysis with the combination of microscopic and spectroscopy technique.

REVIEWERS' COMMENTS:

Reviewer #1 :

The authors answered the questions as much as they can. In my opinion, the results revealing the relationship between electrochemical behaviors and structures in this paper are valuable and will be a good reference in this area, although the critical question about the degree of cation order in the layered structure has not been resolved precisely. The paper can be published in Nature Communications.

Reviewer #2 :

I thank the authors for their responses. As mentioned in the last report, I think the work deserves the publication in NC. The revision provides the clarifications needed and I recommend the publication.

[Optional] If possible, please specify the "covalency" and "hybridization" (page 7).

#3c. As mentioned in the previous report, this manuscript did a much better job on the XAS description of the hybridization pre-edge peaks, compared with the two references quoted in the response, where such hybridization character is misused as evidence of oxygen redox. So I cannot agree with the new addition on page 8 on Fig.3d, because there were not reliable "demonstrations" in those reference papers.

My original question on Fig.3d was to clarify the link between such schematic and SXAS results. I understand the authors try to provide a drawing based on the discussions of covalency variations. But SXAS corresponds to unoccupied states and Fig.3d presents the occupied states, so the last sentence in the figure caption is very unclear.

As an easy solution, the additional text added to page 8 should be removed. The couple words of "observed in O K-edge SXAS" could be just deleted. I am OK with the Fig.3d as a schematic drawing of electron state configurations.

Reviewer #1 :

The authors answered the questions as much as they can. In my opinion, the results revealing the relationship between electrochemical behaviors and structures in this paper are valuable and will be a good reference in this area, although the critical question about the degree of cation order in the layered structure has not been resolved precisely. The paper can be published in Nature Communications.

Response) Thank you for your comment. We added additional explanation about the degree of cation order in revised manuscript (page4) as follows: 'Furthermore, crystallographic parameters for O-MNC and D-MNC were obtained by Rietveld refinement (Supplementary Table. 1 and 2) and discussion in Supplementary Note 1. The results indicate that approximately 3 times more Ni and Co ions have occupied the Li site in D-MNC (total 0.10mol) compared to O-MNC (total 0.03mol).'

Reviewer #2 :

I thank the authors for their responses. As mentioned in the last report, I think the work deserves the publication in NC. The revision provides the clarifications needed and I recommend the publication.

[Optional] If possible, please specify the "covalency" and "hybridization" (page 7).

#3c. As mentioned in the previous report, this manuscript did a much better job on the XAS description of the hybridization pre-edge peaks, compared with the two references quoted in the response, where such hybridization character is misused as evidence of oxygen redox. So I cannot agree with the new addition on page 8 on Fig.3d, because there were not reliable "demonstrations" in those reference papers. My original question on Fig.3d was to clarify the link between such schematic and SXAS results. I understand the authors try to provide a drawing based on the discussions of covalency variations. But SXAS corresponds to unoccupied states and Fig.3d presents the occupied states, so the last sentence in the figure caption is very unclear.

As an easy solution, the additional text added to page 8 should be removed. The couple words of "observed in O K-edge SXAS" could be just deleted. I am OK with the Fig.3d as a schematic drawing of electron state configurations

Response) Thank you for your question. Hybridization is a model, not a phenomenon. However, covalent bonding (covalency) is a phenomenon. Covalent bonding is a type of bonding in which electrons are shared more-or-less locally between atoms. Electrons shared widely between atoms in a crystal called Metallic bonding, electrons "transferred" rather than shared called ionic bonding. Regarding this subject matter, we added following sentence in the revised manuscript (page7). 'Hybridization is a model that modifies atomic orbitals to explain the covalent bonding phenomenon 'covalency''. Furthermore, comments on Fig.3d were reflected in the revised manuscript (page8 and page26).